# Controlling Pepper Mild Mottle Virus (PMMoV) Infection in Pepper Seedlings by Use of Chemically Synthetic Silver Nanoparticles

**DOI:** 10.3390/molecules28010139

**Published:** 2022-12-24

**Authors:** Esam K. F. Elbeshehy, Wael M. Hassan, Areej A. Baeshen

**Affiliations:** 1Department of Biological Sciences, Faculty of Science, University of Jeddah, Jeddah 21959, Saudi Arabia; 2Botany Department, Faculty of Agriculture, Suez Canal University, Ismailia 41522, Egypt; 3Department of Biology, Quwayiyah College of Science and Humanities, Shaqra University, Riyadh 19257, Saudi Arabia

**Keywords:** spherical silver nanoparticles, antiviral activity, PMMoV, pepper

## Abstract

We investigated the roles of different concentrations of chemical synthetic spherical silver nanoparticles (AgNPs) in protecting pepper seedlings of the Mecca region, which were naturally and artificially infected by the pepper mild mottle virus (PMMoV). The virus shows many infection symptoms, including pepper leaf deformation with filiform leaves and severe mosaic symptoms. Our study focused on the antiviral activity of different concentrations of spherical nanoparticles in controlling PMMoV infecting pepper seedlings. PMMoV identification was confirmed via DAS-ELISA using the following antiserum: PMMoV, cucumber mosaic virus (CMV), tobacco mosaic virus (TMV), tomato mosaic virus (ToMV), potato virus Y (PVY), and tomato spotted wilt virus (TSWV). The presence of PMMoV was confirmed using electron microscopy and reverse transcription polymerase chain reaction (RT-PCR). We evaluated the effects of exogenously applied different concentrations of AgNPs on CMV infection rate, infection severity, virus concentration, and the concentrations of photosynthetic pigments chlorophyll a, chlorophyll b, carotenoid content, phenolic compounds, and protein components in virus-infected plant cells that were treated with three different concentration of nanoparticles (200, 300, and 400 µg/L) compared to the positive and negative control.

## 1. Introduction

Among the vegetable crops of great interest and importance is the pepper crop of the Mecca region, which is grown commercially. Recently, pepper plants have been used to evaluate the effects of antimicrobials and viruses [1] and to study the various plant defense mechanisms against viral infections [2,3,4,5,6].

Several viruses infect pepper plants, including PMMoV, CMV, ToMV TMV, PVY, and TSWV [7,8,9,10,11]. The genus Tobamovirus, which includes PMMoV and has a very broad-host range [12], can cause extensive variations in symptoms, such as pepper leaf deformation with filiform leaves and severe mosaic symptoms, which lead to serious yield losses [13]. The viral RNA contains a gene encoding a viral protein that plays a major role in the virus pathogenicity during its invasion of plant cells and leads to the collapse of the plant defense mechanisms, which is a result of their interference and inhibition of cellular plant acids responsible for the resistance induced by plant cells [14].

The use of safer and more environmentally friendly nanoparticle synthesis methods, including biological methods using many living organisms such as primitive nuclei (from bacteria and actinomyces) or eukaryotes (fungi and plants), is gaining worldwide attention [15,16]. Synthetic silver nanoparticles (AgNPs) formed by chemical reduction methods have recently been used as antiviral, antibacterial [17,18,19], antifungal [20], and anticancer agents [21].

In our study, we hypothesized that inhibition of viral infection in infected pepper plant cells by synthetic silver nanoparticles (AgNPs) by chemical reduction methods would be the safest way to curb the viral invasion of plant cells [22]. This modification aimed to assess the effects of different concentrations of silver nanoparticles on virus concentration, infection rate, and infection severity. The effects on phenolic compounds and the total soluble proteins in virus-infected plant cells after treatment with nanoparticles were compared to those in the positive and negative control. [23].

Silver nanoparticles synthesized by the chemical reduction method had a size average of 36.32 nm and played a significant role in the defense mechanism of pepper plants against viral infection. When pepper leaves infected with PMMoV symptoms were treated with three different concentrations of AgNPs in our study, they could limit viral replication inside the leaves and reduce the appearance of infection symptoms on virus-infected plant leaves compared to that in the control [23,24].

## 2. Results and Discussion 

### 2.1. PMMoV Isolation and Identification

Samples of pepper leaves that showed clear symptoms of PMMoV were collected from the different pepper crops in the Mecca region. These symptoms included transparency of the veins (vein clearing) accompanied by severe mosaic formation, reduction in the blade of the leaves (filiform leaves), and small size with distortions of the leaves deformation (see Figure 1a). The samples that tested positive in the ELISA served as the virus source in our study, and artificial mechanical transmission (single local lesion technique) was performed on the leaves of the diagnosed plant Chenopodium giganteum, which recorded the shape of local legions surrounded with little halo edges that would appear on infected leaves. In contrast, the virus was propagated by industrial infection on healthy pepper seedlings, and the symptoms of the infection were recorded after the incubation period of the virus. The symptoms included severe mosaic formation accompanied by deformation of the leaf blade and its reduction. The appearance of symptoms of natural infection in pepper leaves may be owing to the effect of the PMMoV on the internal contents of the cells and the depletion of these contents by the PMMoV, which showed symptoms of external infection corresponding to the same symptoms of the PMMoV in the form of distortions and coloration. The presence of local lesions in the diagnosed host may be attributable to the programmed death of these cells. This result was in agreement with the findings of [22,25,26].

### 2.2. Total RNA Extraction and RT-PCR 

The presence of PMMoV was confirmed via RT-PCR using a primer specific to the virus RNA-dependent RNA polymerase (RdRp) gene amplification. The authors of [27,28] reported that a PCR fragment of an expected size of 830 bp was amplified (see Figure 1b). Naturally, infected pepper plants were tested for PMMoV infection by RT-PCR amplification of the viral RNA, carried out on the total RNA isolated from infected pepper plants using PMMoV-specific primers. Electrophoresis analysis of RT-PCR product showed the 830 bp amplification products of the total RNA, while no products were amplified from the RNA extracted from healthy plants. This result was in agreement with the results of our study [29,30,31].

### 2.3. Electron Microscopy

Microscopic examination of very thin sections of infected and healthy pepper leaves using transmission electron microscopy (TEM) showed a random distribution of the globular virus particles with a laminated aggregate of crystalline particles inside the infected cells (see Figure 1c,d,e). The presence of laminated aggregate of crystalline particles and viral particles associated with abnormalities in the shape and composition of both the nucleus and chloroplasts may be due to the effect of the virus during its replication inside the cell and the depletion of its contents due to such distortions, which was reflected by the appearance of yellow or light green areas on the leaves of pepper plants affected by mosaic [32].

### 2.4. Synthesis and Collection of Spherical AgNPs (avg. 36.32 nm) Using 8.0 mM of Trisodium Citrate Dehydrate (C_6_H_5_O_7_Na_3_)

A surface plasmon resonance (SPR) device at a wavelength of 450 nm, measured using an ultraviolet spectrophotometer, was used to corroborate the configuration of the AgNPs [23,33,34]. The size and potential stability of the AgNPs were analyzed using zeta potential and DLS measurements. The synthesized AgNPs colloid was characterized by XRD (model D5000 Siemens Diffractometer) with 0.15405 nm Cu Kα radiation. The size and morphology of the spherical AgNPs using 8.0 mM of trisodium citrate dehydrate recorded an average of 36.32 nm [23,33,34,35,36].

### 2.5. Activity of AgNPs against PMMoV

Figure 2 and Table 1 show that the treatment of seedlings with different concentrations (200, 300, and 400 µg/L) of AgNPs 24 h after virus inoculation (post-inoculation) resulted in no symptoms appearing in inoculated plants. We also found that treatment with AgNPs during viral inoculation led to a great decrease in the appearance and severity of infection symptoms in the virus-infected leaves compared to symptoms that appeared in seedlings treated 72 h before viral inoculation, which did not affect viral infection or disease severity.

As shown in Figure 2 and Table 1, there was a decrease in infection symptoms when treating virus-infected plants with AgNPs 300 µg/L and AgNPs 400 µg/L compared with AgNPs 200 µg/L. It was observed that pepper seedlings infected with PMMoV and treated with all concentrations of AgNPs 24 h. after inoculation did not show any external symptoms because of the decrease in the virus concentration. Pepper seedlings treated with AgNPs 300 µg/L and AgNPs 400 µg/L 24 h. after viral inoculation (0.451, 0.095, and 0.055) showed infection rates of 44.44, 22.22, and 11.11%, and infection severity of 22.22, 8.33, and 5.56%, respectively, compared with AgNPs 200 µg/L (0.451, 44.44, and 22.22%), suggesting that the approach of viral neutralization by AgNPs occurred during the early viral replication phases. Additionally, increasing the concentration of silver nanoparticles led to a decrease in the accumulation of the virus inside the infected cells, compared to that by treatment with nanoparticles during the pre-viral replication phases, which was as follows: virus concentration (1.156, 1.130, and 0.986), infection severity (100.00, 88.89, and 58.33%), and infection rate (100.00, 88.89, and 58.33%), respectively.

The treatment of seedlings with AgNPs 24 h after a viral infection led to no symptoms of viral infection in plants. We also found that treatment with AgNPs during viral infection led to a great decrease in the appearance and severity of infection symptoms in the virus-infected leaves compared to that in the seedlings treated 72 h after viral inoculation, which did not affect viral infection or disease severity.

This may be due to the inability of the AgNPs to activate induced systemic resistance (ISR) of the plant against PMMoV infection. Moderate reductions in all symptoms were observed when the virus-infected plants were treated with AgNPs simultaneously during the infection in group 2. This result was an agreement with that of [34], which reported that the nanoparticles, especially those synthesized by *B. licheniformis,* showed excellent in vitro antiviral activity against the bean yellow mosaic virus. The significance of the particular antiviral activity is highlighted, given the significant yield reduction in fava bean crops resulting from bean yellow mosaic virus infections in many African countries [37,38,39,40]. This study indicates the ability of silver nanoparticles to prevent the release of viral RNA from the protein coat of the virus. Additionally, blocking the gene responsible for the production of the viral motor protein that allows the virus to move from one cell to another through the filaments of the plasmodesmata shows the effective role of silver nanoparticles in limiting the ability of the virus to spread systemically within the plant cells, and thus limits the appearance of infection symptoms.

### 2.6. Physiological Constraints

#### 2.6.1. Photosynthetic Pigments

In general, the virus has a great ability to affect the breaking down of light pigments inside infected plant cells or indirectly influence the ability of the light pigments to participate in the photosynthesis process, which leads to the emergence of an external infection display such as mosaic symptoms or mottling, and therefore when pepper seedlings were treated with silver nanoparticles after infection with the virus for 24 hours it led to an increase in the content of light pigments in the treated cells, and thus a disappearance in the external symptoms of infection compared to other treatments. Additionally, increasing the concentration of silver nanoparticles led to an increase in the accumulation of the photopigments inside chloroplasts in infected cells, compared to that by treatment with low concentration of nanoparticles during the pre-viral replication phases. Table 2 shows the data that express a clear increase in the photosynthetic pigments (chl a, b, total chl a + b, car, and Chl a + b / Car) in infected plants treated post-inoculation and with inoculation of AgNPs compared to that in the control [33,34,41].

#### 2.6.2. Phenolic Content

In infected pepper leaves inoculated with PMMoV, the total phenolic content was lower than that in healthy leaves. However, infected leaves treated with three concentrations of AgNPs showed a significant increase in phenolic content compared with untreated infected leaves (shown in Table 3). These results are in agreement with those of previous studies [33,42,43,44]. All the plants treated with 300 and 400 µg/L of AgNPs, respectively, showed a higher build-up of phenolic content compared to that of the infected leaves treated with 200 µg/L of AgNPs.

#### 2.6.3. Protein Composition

Infected pepper leaves recorded a high accumulation rate of total soluble protein compared to that in healthy leaves, but infected leaves treated with 200, 300, and 400 µg/L of AgNPs 24 h after PMMoV inoculation were recorded as having inferior build-up of total soluble protein compared to other treatments and the healthy and infected controls. On the other hand, in this study we recorded the lowest decrease of total soluble protein when infected pepper plants were treated with 400 µg/L AgNPs 24 h after inoculation compared with other concentrations of AgNPs (shown in Table 3).

In infected pepper leaves inoculated with PMMoV, the total phenolic content was lower than that in healthy leaves. However, leaves treated with AgNPs showed a significant increase in phenolic content at all treatment concentrations. All plants treated with AgNPs showed a higher build-up of phenolic content compared to that of infected leaves. Infected pepper leaves recorded a high accumulation of total soluble proteins compared to that of healthy leaves, but infected leaves treated with 400 µg/L AgNPs 24 h after PMMoV inoculation were recorded as having an inferior build-up of total soluble proteins compared to 200 µg/L of AgNPs and the control. These results are consistent with those of [38], which mentions that banana plants infected with banana bunchy top virus (BBTV) and treated with 50 ppm AgNPs did not show any external symptoms where the rate of infection was 36%. On the other hand, banana plants treated with 50 ppm AgNPs post-virus inoculation showed non-significant and significant changes in chlorophyll (a and b) and carotenoids, respectively, compared with healthy and nano controls. In contrast, phenol, proline, and oxidative enzymes were significantly increased in all plants treated with 50 ppm AgNPs post-virus inoculation, compared with the healthy control [33,34,41].

## 3. Materials and Methods

Samples of infected pepper *Capsicum annuum L* leaves collected from different crops in the Mecca region generally exhibited symptoms of typical PMMoV infection.

### 3.1. Virus Isolation and Propagation

The presence of pepper mild mottle virus (PMMoV) was confirmed in infected pepper leaf samples collected from different crops in the Mecca region via DAS-ELISA, as explained by ref. [45] using antibodies against PMMoV, CMV, ToMV, TMV, PVY, and TSWV wherein the samples that tested positive in the ELISA served as the PMMoV source. Artificial mechanical transmission of the virus isolated from pepper leaves infected with PMMoV was performed on *Chenopodium giganteum* leaves to purify the virus and record the shape of local legions surrounded with little halo edges that would appear on infected leaves; the virus was then multiplied using artificial mechanical transmission on healthy sweet pepper cv.(Sirtaki) seedlings. Following the incubation period for viral growth and propagation, the symptoms of the infection started appearing and were recorded [26].

### 3.2. Mechanical Transmission of PMMoV

Artificial mechanical transmission was used to transfer the virus artificially from infected pepper leaves to healthy sweet pepper cv.(Sirtaki) seedlings using an abrasive substance to allow the virus to enter the cell without dying.

Infected plant juice was extracted using a phosphate-buffer solution (pH 7) to maintain the stability of the virus during the extraction. Then, the infected seedlings were sprayed with water, and it was during the incubation period of the virus (inside the cells) that the symptoms of the infection started appearing on the newly grown seedlings, which was after approximately 3 weeks [26].

### 3.3. Transmission Electron Microscopy

Ultra-precise sections of the mesophyll tissue of the infected leaves were examined using an electron microscope (JOEL-JEA100 CX.) to detect the presence of inclusion bodies inside the infected cells, as well as the intracellular changes caused by the virus, compared to the healthy sections. Tissues were cut, fixed, and stained according to standard procedures described in ref. [46].

### 3.4. Double Antibody Sandwich ELISA (DAS-ELISA)

The presence of the virus in all infected and healthy plant samples was tested via DAS-ELISA using antibodies specific to PMMoV [47] at a wavelength of 405 nm using an ELISA reader.

### 3.5. Total RNA Extraction and RT-PCR

According to the methods described in ref. [28], specific primers were used for (CP/s:5′-ATGGCATACACAGTTACCAGT-3′and CP/a: 5-′TTAAGGAGTTGTAGCCCACGTA3′, for the one-step PMMoV RNA-dependent RNA polymerase (RdRp) gene amplification, which was similar to RT-PCR.

### 3.6. Synthesis and Collection of Spherical AgNPs (avg. 36.32 nm) Using 8.0 mM of Trisodium Citrate Dehydrate (C_6_H_5_O_7_Na_3_)

A quantity of 80 mL of silver nitrate (AgNo_3_) was added as a starting material to synthesized spherical silver nanoparticles after heating at 60 °C with vigorous stirring to 20 mL (8.0 mM trisodium citrate C_6_H_5_O_7_Na_3_ solution) reducing agent and surfactant pre-heated at 60 °C. The mixture was stirred well for 20 min, then the mixture was cooled to room temperature with continuous stirring [33,35,48,49].

The mechanism of reaction can be expressed as follows: [33].

4Ag^+^ + C_6_H_5_O_7_Na_3_ + 2H_2_O → 4Ag^0^ + C_6_H_5_O_7_H_3_ + 3Na^+^ + H^+^ + O_2_↑

The as-synthesized colloid spherical silver nanoparticles were characterized by XRD (model D5000 Siemens Diffractometer) with 0.15405 nm Cu Kα radiation. The size and morphology of the spherical AgNPs produced were studied by TEM using the methods of [33,35,48,49].

### 3.7. Antiviral Activity of Different Concentrations of AgNPs

The sweet pepper seedlings cv. (Sirtaki) were planted for germination on wet filter paper at room temperature for 2–3 d, and pepper seedlings were transferred to sterilized containers filled with soil and stored in a greenhouse. After 3 weeks of growth, plants of similar sizes were selected and divided into three groups, and each group was divided into eight treatments: T1: (healthy), T2: (NPs-200 µg/L and healthy), T3: (NPs-300 µg/L and healthy), T4: (NPs-400 µg/L and healthy), T5: (infected), T6: (NPs-200 µg/L and infected), T7: (NPs-300 µg/L and infected), T8: (NPs-400 µg/L and infected). Three replicates were allocated for each treatment, and the three groups were divided as follows: the first group was treated with nanoparticles at the time of viral inoculation, the second group was treated with nanoparticles 72 h before the inoculation, and the third group was treated with nanoparticles 24 h after the inoculation [34].

### 3.8. Evaluation of the Changes Resulting from Treatment with AgNPs during Viral Infection

The leaves of healthy and virus-infected pepper plants were treated with AgNPs. After three weeks of treatment, the virus concentration, infection rate, and infection capacity were recorded based on the scale recommended by [36,50]:$$Disease\;severity\;\left(DS\%\right)=\frac{disease\;grade\times number\;of\;plants\;in\;each\;grade}{total\;number\;of\;plants\times highest\;disease\;grade}\;\times \;100$$

#### 3.8.1. Evaluation of the Pigments

The photosynthetic pigments (chlorophyll a, chlorophyll b, and carotenoids) were evaluated as described by [51] in the third leaves of various pepper cultivars 21 dafter inoculation with PMMoV. Concentrations of chlorophyll a, b, and carotenoids in the leaves were calculated as follows:

Chl a = (9.784 × E_662_) − (0.99 × E_644_) = mg/g (F.wt.)

Chl b = (21.426 × E_644_) − (4.56 × E_662_) = mg/g (F.wt.)

Car= (4.695 × E_440.5_) − 0.268 (chla + chlb) = mg/g (F.wt.)

#### 3.8.2. Total Phenolic Content

The total phenolic content was measured using the modified Folin–Ciocalteu method and was calculated as mg/g (DW) using a pyrogallol standard curve, according to [52].

#### 3.8.3. Soluble Protein Content

Total soluble protein content was evaluated in the leaves of infected plants treated with AgNPs and compared with that in the control, according to [53].

### 3.9. Statistical Analyses

The SPSS program was used to perform all statistical analyses of the tables under study, with all numerical results subjected to investigation of disagreement (ANOVA).

## 4. Conclusions

In our study, symptoms included transparency of the veins (vein clearing) accompanied by severe mosaic formation, reduction in the blade of the leaves (filiform appearance), and small size with distortions of the leaves (deformation), which are those of typical infection with PMMoV observed in pepper leaves. The samples that tested positive in the ELISA served as the virus source in our study. The virus was biologically purified after isolation using a single local lesion technique; the local legions were surrounded with little halo edges on *Chenopodium giganteum*. PMMoV RNA-dependent RNA polymerase (RdRp) gene was discovered in infected leaves by RT-PCR when fragments of an expected size of 830 bp were amplified. A study of the ultra-structure showed the presence of laminated aggregate of crystalline particles and viral particles associated with abnormalities in the shape and composition of both the nucleus and chloroplasts. The TEM images confirmed the production of AgNPs at nanoscale; most of them were mono-dispersed with spherical shapes and found to be an average 36.32 nm, confirmed using TEM images. Healthy plants in the experiment were not affected by all concentrations of AgNPs. In infected pepper leaves inoculated with PMMoV, the total phenolic content was lower than that in healthy leaves. However, leaves treated with AgNPs showed a significant increase in phenolic content at all treatment concentrations. All plants treated with AgNPs showed a higher build-up of phenolic content compared to that of infected leaves. Infected pepper leaves recorded a high accumulation of total soluble proteins compared to that of healthy leaves, but infected leaves treated with 400 µg/L of AgNPs 24 h after PMMoV inoculation were recorded as having an inferior build-up of total soluble proteins compared to 200 µg/L of AgNPs and the control.

## Figures and Tables

**Figure 1 molecules-28-00139-f001:**
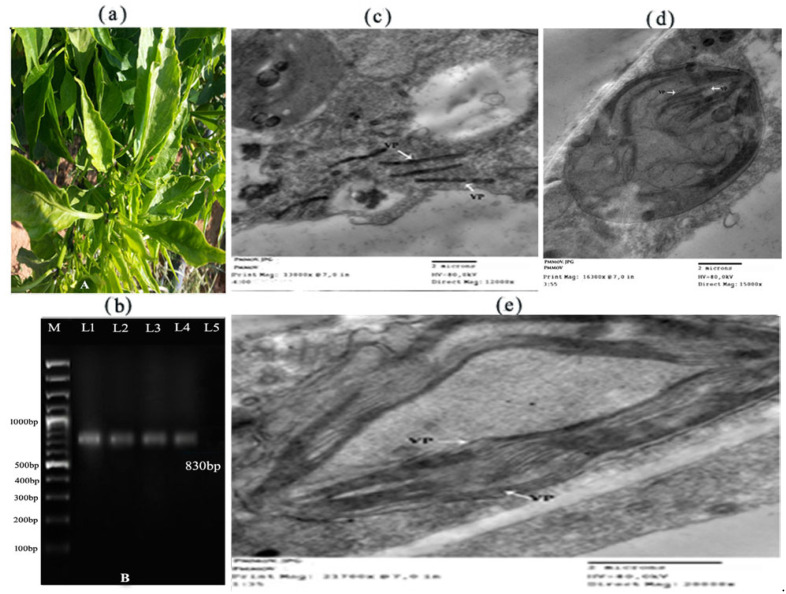
(**a**) Mosaic symptoms of naturally infected pepper were detected positive for PMMoV infection. Vein clearing, filiform leaves, severe mosaic symptoms, and leaf deformation were observed. (**b**) Electropherogram of 1.2% agarose gel showing PCR amplifications from infected tomato with PMMoV. Lane M, DNA ladder marker; lanes 1, 2, 3, and 4 represent PCR-positive infected pepper seedling with PMMoV; lane 5 represents healthy pepper. (**c**–**e**) Microscopic examination of very thin sections of light green area in infected pepper leaves with PMMoV using TEM laminated aggregate of crystalline particles (CP) and Viral particles (VP).

**Figure 2 molecules-28-00139-f002:**
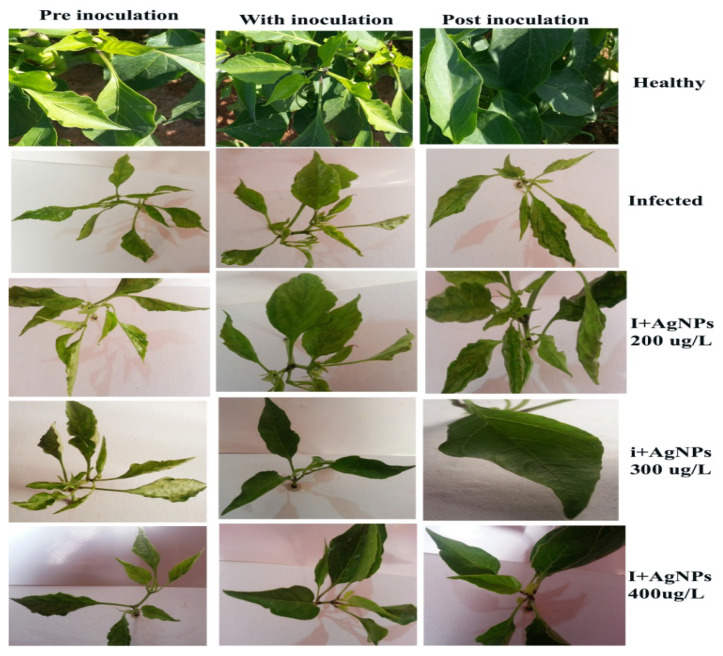
Effect of spherical AgNPs on PMMoV infected and non-infected pepper plants. The nanoparticles were sprayed simultaneously pre-virus inoculation (experimental Group 1); with inoculation (Group 2), or after inoculation (Group 3). Vein clearing, filiform leaves, severe mosaic symptoms, and leaf deformation were observed in the case of infected leaves. These symptoms were reduced or completely disappeared when plants were treated with silver nanoparticles.

**Table 1 molecules-28-00139-t001:** Effect of different concentrations of silver nanoparticles on estimated percentage of infection and disease severity in sweet pepper, cv.(Sirtaki) in the presence PMMoV under greenhouse conditions.

Groups	Treatments	Virus Concentration	* Virus Infectivity			Percentage of Infection	Percentage of Disease Severity (DS %) (*)
			R_1_	R_2_	R_3_		
**Pre-inoculation**	**Negative control**	0.034 (−)	0/3	**0/3**	**0/3**	**0.00%**	**0.00%** **(0)**
	**Positive control**	1.164 (+)	3/3	3/3	3/3	100%	100% (4)
	**Infected and AgNPs (200 µg/L)**	1.156	3/3	3/3	3/3	100%	100% (4)
	**Infected and AgNPs (300 µg/L)**	1.130	3/3	2/3	3/3	88.89%	88.89% (4)
	**Infected and AgNPs (400 µg/L)**	0.986	3/3	2/3	2/3	66.67%	58.33% (3)
**With inoculation**	**Infected and AgNPs (200 µg/L)**	0.897	2/3	2/3	3/3	77.78%	58.33% (3)
	**Infected and AgNPs (300 µg/L)**	0.312	1/3	1/3	1/3	33.33%	16.67% (2)
	**Infected and AgNPs (400 µg/L)**	0.121	1/3	0/3	1/3	22.22%	5.56% (1)
**Post-inoculation**	**Infected and AgNPs (200 µg/L)**	0.451	1/3	2/3	1/3	44.44%	22.22% (2)
	**Infected and AgNPs (300 µg/L)**	0.095	1/3	0/3	1/3	22.22%	8.33% (1)
	**Infected and AgNPs (400 µg/L)**	0.055	1/3	0/3	0/3	11.11%	5.56% (1)

(*)*:* Disease grade.

**Table 2 molecules-28-00139-t002:** Changes of photosynthetic pigments in control and PMMoV infected pepper leaves under effect of different silver nanoparticle concentrations.

Groups	Treatments	*Chl a* (mg/g Fresh Weight)	*Chl b* (mg/g Fresh Weight)	*Ch* a + b	*Car.* (mg/g Fresh Weight)	*Chla+b / Car.*
**Pre-inoculation**	**Negative control**	1.099	0.396	1.495	0.508	2.943
	**Positive control**	0.595	0.232	0.827	0.253	3.269
	**Infected and AgNPs (200 µg/L)**	0.596	0.221	0.817	0.276	2.960
	**Infected and AgNPs (300 µg/L)**	0.616	0.244	0.860	0.288	2.986
	**Infected and AgNPs (400 µg/L)**	0.825	0.256	1.081	0.297	3.639
**With inoculation**	**Infected and AgNPs (200 µg/L)**	0.912	0.241	1.153	0.412	2.798
	**Infected and AgNPs (300 µg/L)**	1.024	0.311	1.335	0.427	3.127
	**Infected and AgNPs (400 µg/L)**	1.052	0.323	1.375	0.436	3.154
**Post-inoculation**	**Infected and AgNPs (200 µg/L)**	1.037	0.367	1.404	0.412	3.408
	**Infected and AgNPs (300 µg/L)**	1.041	0.378	1.419	0.495	2.867
	**Infected and AgNPs (400 µg/L)**	1.085	0.395	1.477	0.501	2.948

**Table 3 molecules-28-00139-t003:** Changes of total phenols (mg/g dry weight) and soluble proteins (mg/ g fresh weight) in control and infected pepper leaves with PMMoV under effect of different silver nanoparticle concentrations.

Treatments	Pre-Inoculation		With Inoculation		Post-Inoculation	
	Total Phenols (mg/g dw)	Soluble Proteins (mg/g fw)	Total Phenols (mg/g dw)	Soluble Proteins (mg/g fw)	Soluble Proteins (mg/g fw)	Total Phenols (mg/g dw)
**Negative control**	0.582	29.84	0.579	29.77	0.580	0.580
**Positive control**	0.323	45.81	0.365	45.99	0.377	0.377
**Infected and AgNPs (200 µg/L)**	0.311	43.61	0.399	41.29	0.385	0.385
**Infected and AgNPs (300 µg/L)**	0.347	36.66	0.489	32.26	0.530	0.530
**Infected and AgNPs (400 µg/L)**	0.352	38.52	0.497	31.23	0.573	0.573

## Data Availability

The data presented in this study are available on request from the corresponding author.

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
