# Peer review of "Controlling Pepper Mild Mottle Virus (PMMoV) Infection in Pepper Seedlings by Use of Chemically Synthetic Silver Nanoparticles"

_molecules, 2022, doi:10.3390/molecules28010139_

Round 1
Reviewer 1 Report
The research paper as a whole is of great importance in the applied field, as it highlighted the importance of silver nanoparticles in limiting the spread of the virus PMMoV on pepper seedlings. Therefore the research is considered the cornerstone for applying silver nanoparticles to many plant viruses that infect strategic crops and have an Economic impact in the region under study.
Please from the research author:
1- Highlighting the mode of action of silver nanoparticles on the virus inside infected cells and explaining how they affect it.
2- Updating some old references in the research
3- In Table 1, the column representing the percentage of injury severity, clarifying the meanings of the numbers in the brackets.
Author Response
We would like to thank the reviewers for the comments which are extremely helpful to me as a corresponding author to modify this MS in order to help us to publish our MS in high quality. With the submission of this revised manuscript I make most of the reviewers comments, I highlight the changes within the document by red color. Also, I attached file names (Respond to Decision letter and reviewer (s) comments)
Thanks in advance for your time and consideration.
Dr. Esam K.F. Elbeshehy ( Corresponding Author)
Professor in Plant Virology
Biological Science Department
Faculty of Science, Jeddah
University - Jeddah - KSA
Tel. +966 532895704 ; E-mail: eyossef@uj.edu.sa ; esamelbeshehy@yahoo.com

Reviewer 2 Report
Thanks for submitting an article to this journal.
It would help if you revised the entire article.
Correct spelling mistakes.
Correct grammar mistakes.
Aboutnanoparticles, Reinforce the introduction and discussion sections and refer to other related references such as:
Al Otraqchi KI, Darogha SN, Ali BA. Serum levels of immunoglobulin and complement in UTI of patients caused by Proteus mirabilis and using AgNPs as antiswarming. Cellular and Molecular Biology. 2021 Nov 25;67(3):11-23. https://doi.org/10.14715/cmb/2021.67.3.3.
Alavi, M., Adulrahman, N. A., Haleem, A.A., Al-Râwanduzi, A.D.H., Khusro, A., Abdelgawad, M. A., Ghoneim, M. M., Batiha, G. E.-S., Kahrizi, D., Martinez, F., & Koirala, N. (2022). Nanoformulations of curcumin and quercetin with silver nanoparticles for inactivation of bacteria. Cellular and Molecular Biology, 67(5), 151–156. https://doi.org/10.14715/cmb/2021.67.5.21.
Cortés H, Reyes-Hernández OD, Gonzalez-Torres M, Vizcaino-Dorado PA, Del Prado-Audelo ML, Alcalá-Alcalá S, Sharifi-Rad J, Figueroa-González G, González-Del Carmen M, Florán B, Leyva-Gómez G. Curcumin for parkinson´ s disease: potential therapeutic effects, molecular mechanisms, and nanoformulations to enhance its efficacy. Cellular and Molecular Biology. 2021 Jan 31;67(1):101-5. https://doi.org/10.14715/cmb/2021.67.1.15.
Feng X, Zhu S, Yan Z, Wang C, Tong W, Xu W. PDGFRB as a potential therapeutic target of ankylosing spondylitis: validation following bioinformatics analysis. Cellular and Molecular Biology. 2020 Sep 30;66(6):127-34. https://doi.org/10.14715/cmb/2020.66.6.23.
Ibrahem KH, Ali FA, Sorchee SM. Biosynthesis and characterization with antimicrobial activity of TiO2 nanoparticles using probiotic Bifidobacterium bifidum. Cellular and Molecular Biology. 2020 Oct 31;66(7):111-7. https://doi.org/10.14715/cmb/2020.66.7.17.
Leroux MM, Doumandji Z, Chezeau L, Hocquel R, Ferrari L, Joubert O, Rihn P, Rihn BH. Validation of an air/liquid interface device for TiO2 nanoparticle toxicity assessment on NR8383 cells: preliminary results. Cellular and Molecular Biology. 2020 Sep 30;66(6):112-6. https://doi.org/10.14715/cmb/2020.66.6.20.
Mohammed AK, Salh KK, Ali FA. ZnO, TiO2 and Ag nanoparticles impact against some species of pathogenic bacteria and yeast. Cellular and Molecular Biology. 2021 Nov 25;67(3):24-34. https://doi.org/10.14715/cmb/2021.67.3.4.
Sami A, Naqvi SS, Qayyum M, Rao AR, Sabitaliyevich UY, Ahmad MS. Calcium based siRNA coating: a novel approach for knockdown of HER2 gene in MCF-7 cells using gold nanoparticles. Cellular and Molecular Biology. 2020 Sep 30;66(6):105-11. https://doi.org/10.14715/cmb/2020.66.6.19.
Author Response
Respond to the decision letter and reviewers' comments
|
No |
Reviewer 1 Comments |
Answer to the Comments or Author Action |
|
1 |
Highlighting the mode of action of silver nanoparticles on the virus inside infected cells and explaining how they affect it. |
Add the paragraph has been highlighting the mode of action of silver nanoparticles on the virus inside infected cells and explaining how they affect it |
|
2 |
Updating some old references in the research |
Updating many references
|
|
3 |
In Table 1, the column representing the percentage of injury severity clarifies the meanings of the numbers in the brackets. |
Clarifying |
|
No |
Reviewer 2 Comments |
Answer to the Comments or Author Action |
|
1 |
|
Corrected |
|
2 |
Correct grammar mistakes. |
Corrected |
|
3 |
About nanoparticles, Reinforce the introduction and discussion sections and refer to other related references |
Five types of research have been added to the body of the research and also in the references among the research recommended by the reviewer |
